

# Spatial heterogeneity of soil properties in planted mixed forests in the rocky desertification areas of the Wuling Mountain

Ziqian Pan*, Yanyan Dong*, Gongxiu He, Tongtong Guo and Ninghua Zhu

College of Forestry, Central South University of Forestry and Technology, Changsha, Hunan, China
* These authors contributed equally to this work.

Corresponding author
Gongxiu He, hegongxiu@163.com

## ABSTRACT

In this study, experiments were conducted on soil samples collected from depths of 0–15 cm, 15–30 cm, and 30–50 cm at the National Long-term Scientific Research Base for the Comprehensive Management of Rocky Desertification in the Wuling Mountains. The aim was to determine the physicochemical indexes and explore the nature and spatial heterogeneity of the soil of the planted mixed forests within the rocky desertification area of the Wuling Mountain. Various analytical methods were employed, including descriptive statistical analysis, correlation analysis, analysis of variance, principal component analysis, spatial interpolation analysis, and kriging interpolation, to fit the optimal model of the semi-variance function of soil physicochemical properties and analyze the model's parameters. The results indicated that soil physical and chemical properties varied with depth and were generally correlated. The relationship between soil organic matter and total nitrogen content was the closest. Additionally, there was a certain degree of correlation between soils at different depths in the vertical profile, generally the correlation between layer B (15–30 cm) and layer C (30–50 cm) > that between layer A (0–15 cm) and layer B (15–30 cm) > that between layer A (0–15 cm) and layer C (30–50 cm). The weighting coefficients of the principal components of soil physicochemical properties indicated that soil organic matter, nitrogen, phosphorus, potassium, pH, total porosity, and capillary porosity are key factors in the soil properties of karst desertification areas. The spatial variability of soil physicochemical properties at different depths ranged from 21.91 to 87.59 m, and the abutment ratio (Co/Co+C) of these properties ranged from 12.99% to 89.53%. Using kriging interpolation in ArcGIS, the spatial distribution pattern of soil physical and chemical properties was mapped, revealing that these indicators were distributed with heterogeneous patches of various sizes and shapes. Therefore, the degree of rocky desertification significantly influences the spatial distribution pattern of soil physical and chemical properties.

# INTRODUCTION

China's karst landforms are mainly distributed in Guizhou, Hunan, Sichuan, Guangxi, and other southwestern regions, covering about 5.35% of the national land area. In recent years, the issue of rocky desertification in these areas has become increasingly severe, resulting in fragile ecosystems that negatively impact local productivity and economic development (*Yang, 2012*; *Gao et al., 2021*). Rocky desertification not only impacts the natural environment but also poses a threat to local social development (*Williams, 1993*; *Wu et al., 2023*). Soil plays a crucial role in the entire ecosystem, and its quality directly relates to ecological balance and sustainable development (*Coleman, Crossley & Hendrix, 2004*; *Odum & Barrett, 2005*). However, due to natural conditions such as geomorphology, soil quality in some areas has begun to decline, manifested by the intensification of soil erosion (*Wu et al., 2022*; *Pu et al., 2022*). As the degree of soil erosion further deepened, it led to the continuous development of rocky desertification in karst areas, resulting in a drastic decline in soil quality (*Voroney, 2007*). The same soil texture type at the same latitude and longitude, soil properties in space will also show significant differences (*Jenny, 1984*; *Wiens, 1989*; *Levin, 1992*), called soil spatial heterogeneity. The effectiveness of soil nutrient elements and soil spatial heterogeneity are important factors for maintaining productivity and conserving biodiversity in the whole ecosystem (*Ettema & Wardle, 2002*; *Wang, Li & Li, 2003*), and the parent soil and local climate largely determine the spatial heterogeneity of soil properties (*Brady & Weil, 1952*). Therefore, understanding the changes in soil physical and chemical properties and their spatial heterogeneity not only aids in the scientific management of rocky desertification areas but also provides a theoretical basis for ecological restoration and sustainable development.

Research on the causes of rocky desertification has yielded a great deal of results in China. Most of the studies have shown that natural and anthropogenic factors lead to the phenomenon of rocky desertification, and this has been generally recognized by scholars (*Jing et al., 2014*; *Zhang, Huang & Zhou, 2020*). Experts and scholars also classify the grades of rocky desertification according to indicators such as the vegetation coverage rate, rock exposure rate, land use type, and soil erosion degree in rocky desertification areas (*Li et al., 2004*; *Li, Dong & Wang, 2007*; *Wang et al., 2007*; *Liu & Wang, 2009*; *Cheng, 2009*). Scholars have studied different aspects of soil physicochemical properties in rocky desertification areas, mainly including soil physicochemical properties and their correlation under different rocky desertification degrees, variation characteristics of a certain nutrient element in rocky desertification areas, the influence of different land use modes on soil physicochemical properties, characteristics of soil physicochemical properties under different slopes, and the response of rocky desertification succession to soil physicochemical properties, *etc*., (*Li et al., 2006*, *2008a*; *Li, Dong & Wang, 2007*; *Si et al., 2009*; *Wang, 2009*; *Pang et al., 2016*; *Jing et al., 2016*; *Tian, 2016*; *Yan et al., 2016*), These findings provide a solid theoretical foundation for the effective management of

rocky desertification areas. Researchers regard the spatial dependency of soil characteristics as an interfering factor throughout the study. so they will apply various treatments to counteract this spatial autocorrelation (*Legendre, 1993*; *Webster, 2008*). Therefore, early studies were unable to effectively explore soil spatial autocorrelation variables at different spatial scales due to the lack of suitable statistical methods for analysis (*Matheron, 1963*; *Mcgrath, Zhang & Carton, 2004*; *Chaoyang, Cheng & Linsheng, 2009*). As a result of ongoing research in geostatistics, scholars such as Aersoy, Necat Agca, and David Mcgrath (*Campbell, 1978*; *Trangmar, Yost & Uehara, 1986*; *Luo et al., 2007*; *Neto, Junior & Oliveira, 2020*) applied geostatistical analysis methods to the study of soil spatial variability and achieved remarkable results. *Campbell (1978)* first applied geostatistical spatial analysis methods to the study of spatial heterogeneity characterization of soil physicochemical properties Campbell first applied geostatistical spatial analysis methods to the study of spatial heterogeneity of soil physical and chemical properties, and the study of soil physical and chemical properties began to transform from qualitative research to quantitative research (*Ingestad, 1981*; *Webster, 1985*; *Condit, 1995*; *Myers et al., 2000*).

Due to the spatial heterogeneity of physicochemical properties in complex soil ecosystems at different scales, which makes it difficult for many experts and scholars to study (*Aerts & Chapin, 1999*; *Elser et al., 2007*), the research is still in its preliminary stages (*Heuvelink & Webster, 2001*). At present, the main research focuses on the theoretical connotation, natural distribution, formation mechanism, occurrence and development of rocky desertification. There are fewer studies on the physicochemical properties and related aspects of rocky desertification soils (*Scanlon & Thrailkill, 1987*; *Sullivan, 2014*; *Zhang, 2014*; *Vallejos et al., 2015*; *Romanazzi, Gentile & Polemio, 2015*).

The soil in the sample area of Wuling Mountain National Rocky Desertification Comprehensive Treatment Research Base is limestone yellow-red soil. The soil exhibits poor continuity and is unevenly distributed in karst gullies and fissures. In this study, In order to further understand the spatial properties of soil physical and chemical properties in rocky desertification areas, and to provide a basis for the optimization and adjustment of the forestry structure in rocky desertification areas as well as for the restoration and reconstruction of their ecosystems, samples were collected from soil layers at depths of 0–15 cm, 15–30 cm, and 30–50 cm to reveal the variations in physical and chemical properties across different soil layers. This helps in understanding the spatial heterogeneity of soil and its impact on ecosystem productivity, thereby exploring the characteristics of artificial mixed forest soils in the rocky desertification area of Wuling Mountain. The soil samples were used to determine the physicochemical indexes of the soil samples, in order to investigate the soil properties and spatial heterogeneity of the artificial mixed forests in the Wuling Mountain.

## MATERIALS AND METHODS

### Study site

The National Long-term Scientific Research Base for Comprehensive Management of Rocky Desertification in the Wuling Mountain is located in Qingping Town, Yongshun County, Xiangxi Autonomous Prefecture, Hunan Province, with longitude of 110°13′

**Table 1  General situation of sample plot.**

| Sample point number | Bareness Degree of rock (%) | Altitude (h/m) | Slope ($\alpha(°)$) | Density ($\rho$/plants/hm$^2$) | Average breast Diameter (d/cm) | Average Height (h/m) |
|---|---|---|---|---|---|---|
| $A_1$–$A_{10}$ | 5.5 | 462 | 7.3 | 0.16 | 16.12 | 14.62 |
| $B_1$–$B_{10}$ | 12.5 | 465 | 5 | 0.13 | 16.64 | 13.28 |
| $C_1$–$C_{10}$ | 30.5 | 467 | 13 | 0.09 | 17.59 | 15.05 |
| $D_1$–$D_{10}$ | 44.5 | 470 | 16.1 | 0.08 | 20.38 | 18.45 |
| $E_1$–$E_{10}$ | 45 | 473 | 12.8 | 0.07 | 19 | 14.63 |
| $F_1$–$F_{10}$ | 60 | 475 | 9.5 | 0.05 | 20.8 | 11.71 |
| $G_1$–$G_{10}$ | 76 | 476 | 19.5 | 0.04 | 17.76 | 14.6 |
| $H_1$–$H_{10}$ | 32.5 | 476 | 12.7 | 0.42 | 19.42 | 16.07 |
| $I_1$–$I_{10}$ | 37.5 | 476 | 8.5 | 0.09 | 17.06 | 14.02 |
| $J_1$–$J_{10}$ | 20 | 476 | 17.2 | 0.11 | 16.22 | 13.29 |

**Note:**
Sample point number, Represents the identification number for each sample point; Bareness (degree of rock %), Indicates the percentage of exposed rock within the sample point; Altitude (h/m), Altitude of the sample point in meters; Slope ($\alpha°$), Slope of the sample point in degrees; Density ($\rho$/plants/hm$^2$), Plant density per hectare at the sample point; Average breast diameter (d/cm), Average breast height diameter of trees at the sample point, in centimeters; Average height (h/m), Average height of trees at the sample point, in meters.

40.296″ East and latitude of 29°3′21.59″ North, which is in the center of the Wuling Mountain. The highest elevation is 820 m and the lowest elevation is 320 m. The parent rock is limestone, which belongs to the area of heavy rocky desertification. The soil is Zhongshan limestone yellow-red soil, Zhongshan limestone red-yellow soil and Zhongshan limestone yellow soil. Influenced by the subtropical monsoon climate, the study sample site has abundant rainfall and favorable light and heat conditions, with an average annual sunshine hours of 1,306 h, a subtropical climate, an average annual temperature of 16°, and an annual rainfall of about 1,300 mm (*Nie, 2012*).

## Sample collection

This study is located in Hunan Province Wuling Mountain rocky desertification comprehensive treatment of national long-term scientific research base, according to the distribution of forest sample plots (pure forest, mixed forest) in the scientific research base, survey the entire plantation afforestation lot, choose to represent the establishment of one hectare in the plantation mixed forests in the establishment of a permanent standard sample plot, the basic overview of the standard sample plots as shown in Table 1. The 10 m × 10 m small sample squares evenly distributed in the plot were numbered, the selection of sample plots aims to avoid proximity to the edges of the sampling area to reduce the influence of edge effects. The numbering of the sample squares is shown in Fig. 1. Soil sampling was carried out in each small sample square.

Field sampling was conducted in June-July 2023 and sampling was conducted according to the sample plot setup plan described above, with each sampling site marked. Sampling tools were mainly augers and ring knife samplers. Field sampling was carried out one by one according to the set sampling points. Sampling was carried out in three layers of 0–15 cm (A), 15–30 cm (B), and 30–50 cm (C), and the ring knife was taken in the middle of each layer (*Bhavya et al., 2018*). Collect approximately 500 g of soil from each layer and

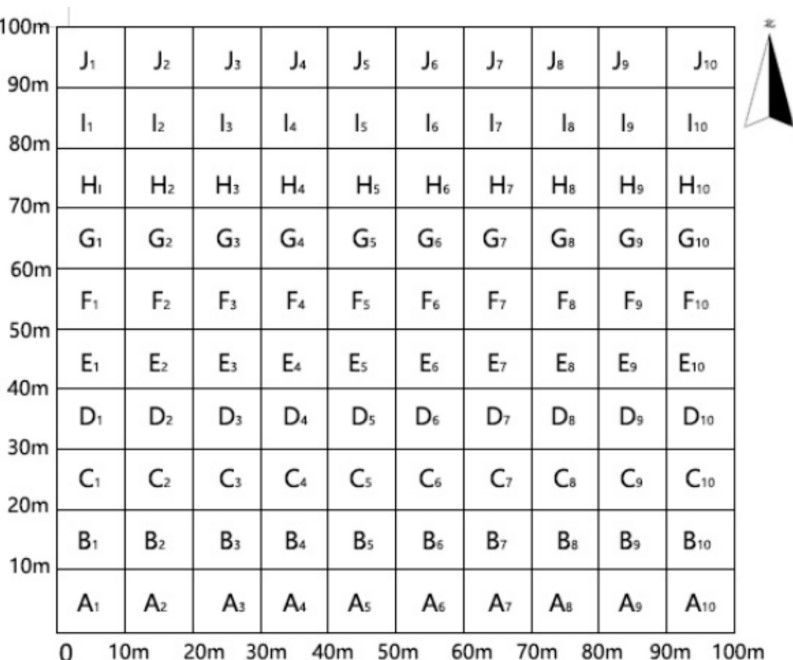

**Figure 1 Plot_number_diagram.** The spatial distribution of sample points within the study area. Each cell corresponds to a sample point location, and the numbers within the cells correspond to the sample point numbers listed in Table 1.

place it in a ziplock bag, promptly removing any debris such as plant roots, stones, dead branches, and leaves. The natural environment of the sampling site was carefully recorded, including the presence of fallen trees and dead standing trees at the edge of the sampling site. All soil samples were transported to the laboratory and then dry in a natural state.

## Determination of soil properties

This study measured eight soil physical properties and five soil chemical properties. Following the national forestry industry standard, "Determination of Soil Moisture—Physical Properties of Forest Soil" (*National Standards of the People's Republic of China, 1999*), the soil water content was determined using the drying method, while the bulk density, capillary porosity, non-capillary porosity, total porosity, capillary water-holding capacity, maximum water-holding capacity, and minimum water-holding capacity were measured using the ring-knife water immersion method (*Institute of Soil Science, Chinese Academy of Sciences, 1978*). The soil pH was determined with a soil: water ratio of 1: 2.5 (w/v) measured using a pH meter (PHS-3C; INESA Scientific Instrument Co., Ltd., Shanghai, China) (*Kader et al., 2015*). The soil organic matter content was measured using the potassium dichromate external heating method (*Bao, Yang & Xu, 2000*), the total nitrogen (TN) content was determined using a carbon-nitrogen element analyzer, total phosphorus (TP) was measured using the sodium hydroxide fusion-molybdenum-antimony anti-spectrophotometric method, and total potassium (TK) was determined using the sodium hydroxide fusion-flame photometry method. (*Richard & Donald, 1996*; *De Rooij, 2004*).

## Data processing and analysis

The raw data were statistically analyzed and calculated using Excel, and the data of thirteen soil physical and chemical properties were subjected to descriptive statistical analysis, inter-indicator correlation analysis, one-way analysis of variance (ANOVA) and LSD *post-hoc* tests using IBM SPSS Statistics 26 software (SPSS Inc., Chicago, IL, USA). ArcGIS Geostatistical Wizard was used to analyze the spatial heterogeneity of soil physical and chemical properties and to fit the semi-variance function model of soil physical and chemical properties. At the same time, spatial ordinary kriging interpolation was performed to map the spatial distribution pattern of soil physical and chemical properties. Kriging interpolation takes into account the spatial autocorrelation of the data, providing high accuracy and applicability (*Miao & Wang, 2024*). The semi-variance function γ(h) in geostatistics is only related to the sample spacing h in this case where the mean of the random function is stable and the variance exists and is finite, and it can be defined as half of the increment of the random function Z(x):

$$\gamma(h) = 1/2N(h) \sum [z(si) - z(si + h)]^2.$$

Notes: γ(h) is the semi-variance statistic: N(h) is the number of pairs of points spaced at distance h; Z(si) is the observed value of the variable under consideration at Si; Z(si + h) is the observed value of the same variable at distance h from point si.

# RESULTS

## Descriptive statistical analysis of soil physical and chemical properties

The physical and chemical properties of the soil change consistently with increasing depth. As depth increases, soil bulk density rises gradually, while soil moisture content, porosity, and water-holding capacity generally decrease. The levels of organic matter, TN, and TP decline significantly, particularly organic matter, whereas TK increases with depth.

According to the different physical and chemical properties of the soils, the degree of variability in soil physical and chemical properties also differed, with the coefficient of variation ranging from 0.1 to 0.74. The degree of variability of soil bulk density and pH were very weak, and the degree of variability of soil capillary porosity, total soil porosity, soil capillary water holding capacity, maximum water holding capacity, TN, TP was moderately varied, and the degree of variability in soil moisture content, soil non-capillary porosity, and organic matter was relatively high. The different coefficients of variation of the soils indicate that the physical and chemical properties of the soils differ in vertical space.

## Statistical analysis of correlation between soil physical and chemical properties

As shown in (Table 2), there was a highly significant negative correlation ($P < 0.01$) between soil bulk density and soil moisture content, capillary water holding capacity, maximum water holding capacity, minimum water holding capacity, and there was a

**Table 2 Correlation analysis of soil physical and chemical properties.**

| Indicators | X₁ | X₂ | X₃ | X₄ | X₅ | X₆ | X₇ | X₈ | X₉ | X₁₀ | X₁₁ | X₁₂ | X₁₃ |
|---|---|---|---|---|---|---|---|---|---|---|---|---|---|
| $X_1$ | 1 | | | | | | | | | | | | |
| $X_2$ | −0.317 | 1 | | | | | | | | | | | |
| $X_3$ | −0.578 | 0.849** | 1 | | | | | | | | | | |
| $X_4$ | −0.666* | 0.843** | 0.936** | 1 | | | | | | | | | |
| $X_5$ | −0.511 | 0.858** | 0.961** | 0.9** | 1 | | | | | | | | |
| $X_6$ | −0.306 | 0.227 | 0.053 | 0.391 | 0.057 | 1 | | | | | | | |
| $X_7$ | −0.249 | 0.87** | 0.925** | 0.803** | 0.909** | −0.088 | 1 | | | | | | |
| $X_8$ | −0.361 | 0.919** | 0.904** | 0.923** | 0.889** | 0.313 | 0.917** | 1 | | | | | |
| $X_9$ | −0.095 | 0.202 | 0.135 | 0.16 | 0.128 | 0.127 | 0.139 | 0.181 | 1 | | | | |
| $X_{10}$ | −0.026 | 0.071 | 0.065 | 0.141 | 0.037 | −0.025 | 0.067 | 0.054 | −0.012 | 1 | | | |
| $X_{11}$ | −0.02 | 0.051 | 0.043 | 0.022 | 0.009 | −0.025 | 0.042 | 0.032 | 0.076 | 0.947** | 1 | | |
| $X_{12}$ | −0.098 | −0.097 | 0.008 | 0.014 | −0.037 | −0.014 | −0.055 | −0.046 | −0.15 | 0.538 | 0.542 | 1 | |
| $X_{13}$ | 0.046 | 0.15 | 0.007 | 0.076 | 0.082 | −0.053 | 0.117 | 0.13 | 0.684* | −0.073 | −0.012 | −0.204 | 1 |

**Notes:**

X1, Bulk density; X2, Moisture content; X3, Capillary water holding capacity; X4, Maximum water holding capacity; X5, Minimum water holding capacity; X6, Capillary porosity; X7, Non-capillary porosity; X8, Total porosity; X9, pH; X10, Organic matter; X11, Total Nitrogen; X12, Total Phosphorus; X13, Total Potassium.
* Significant correlation at the 0.05 level (two-sided).
** Significant correlation at the 0.01 level (two-sided).

significant negative correlation ($P < 0.05$) with capillary porosity and total porosity, which indicated that the higher the soil density and the tighter the structure were, the lower the porosity was, and the moisture content of soil was reduced accordingly. There was a significant negative correlation ($P < 0.01$) between soil pH and soil TP content, and a significant positive correlation ($P < 0.01$) with TK content. This was mainly due to the fact that the whole phosphorus mainly existed in the form of inorganic phosphate, the higher TP content, the smaller the pH value indicated by phosphate ions; TK mainly existed in the form of alkaline mineral potassium salts, the higher the whole potassium content, the larger the pH value indicated by potassium ions. Soil organic matter content and soil TN and TP content had a significant positive correlation ($P < 0.01$), in which the positive correlation coefficient between soil organic matter and soil total nitrogen was the strongest, with a correlation coefficient of 0.95. The study showed that the soil organic matter and total nitrogen content were most closely related. The correlation coefficient was 0.95, the results indicate that soil organic matter is most closely related to total nitrogen content, suggesting that organic matter plays a critical role in nitrogen cycling. The negative correlation between soil organic matter and total potassium was significant ($P < 0.05$), which was probably related to the adsorption of potassium ions by the organic matter, and weakened the fixation of potassium by potassium-containing minerals.

Soils at different vertical profile depths also showed some correlations, generally the correlation between layer B (15–30 cm) and layer C (30–50 cm) > that between layer A (0–15 cm) and layer B (15–30 cm) > that between layer A (0–15 cm) and layer C (30–50 cm).

**Table 3 Principal component eigenvalue and variance contribution rate.**

| Component | Initial eigenvalues | | | Extract sum of squares loading | | |
|---|---|---|---|---|---|---|
| | Eigenvalues | Variance contribution (%) | Cumulative variance contribution (%) | Eigenvalues | Variance contribution (%) | Cumulative variance contribution (%) |
| 1 | 6.829 | 38.313 | 38.313 | 6.829 | 38.313 | 38.313 |
| 2 | 2.406 | 32.667 | 70.908 | 2.406 | 32.667 | 70.908 |
| 3 | 1.668 | 14.749 | 85.729 | 1.668 | 14.749 | 85.729 |
| 4 | 1.293 | 9.882 | 95.557 | 1.293 | 9.882 | 95.557 |
| 5 | 0.76 | 1.633 | 96.19 | | | |
| 6 | 0.51 | 3.644 | 97.112 | | | |
| 7 | 0.301 | 0.922 | 98.339 | | | |
| 8 | 0.108 | 0.77 | 99.109 | | | |
| 9 | 0.059 | 0.421 | 99.531 | | | |
| 10 | 0.048 | 0.341 | 99.871 | | | |
| 11 | 0.015 | 0.106 | 99.977 | | | |
| 12 | 0.002 | 0.015 | 99.992 | | | |
| 13 | 0.001 | 0.008 | 100 | | | |

Note:
Component, the principal component number; Initial eigenvalues, Eigenvalues of the principal components before rotation; Variance Contribution (%), the percentage of the total variance explained by each principal component; Cumulative Variance Contribution (%), the cumulative percentage of the total variance explained by the principal components up to that point; Extract sum of squares loading, Eigenvalues and variance contribution percentages after extracting the sum of squares loading.

## Principal component analysis of soil physical and chemical properties

Dimensionality reduction and factor analysis were performed on the soil physicochemical property indicators using SPSS, and components with eigenvalues greater than 1 were retained as principal components. In this study, 13 soil physicochemical property indicators were subjected to factor analysis and principal component extraction (Table 3). The eigenvalue of the first principal component was 6.83 with a variance contribution of 38.31%, the eigenvalue of the second principal component was 2.40 with a variance contribution of 32.67%, the eigenvalue of the third principal component was 1.67 with a variance contribution of 14.75%, the eigenvalue of the fourth principal component was 1.29 and variance contribution rate of 9.88%. The cumulative variance contribution rate of the four principal components is 95.56%, the variation in soil physicochemical properties is mainly concentrated in these four principal components. And the four principal components can represent the overall information of soil characteristics of rocky desertification in the study area, explaining 95.56% of the differences in soil characteristics of rocky desertification in the study area.

The weight coefficients of the principal components of the soil physicochemical property indicators can be concluded that soil organic matter, nitrogen, phosphorus, potassium, pH, total porosity, capillary porosity are the key factors of soil physicochemical properties within the karst rocky desertification area, which play an important role in the improvement of soil nutrient cycling, and in the formation and stabilization of soil structure (Table 4).

**Table 4 Principal component weight coefficient of soil physical and chemical properties.**

| Soil physical and chemical indicators | Weighting factor | | | |
|---|---|---|---|---|
| | Principal component 1 | Principal component 2 | Principal component 3 | Principal component 4 |
| X1 | 0.509 | −0.452 | 0.311 | −0.661 |
| X2 | −0.345 | −0.465 | 0.548 | 0.211 |
| X3 | −0.593 | 0.687 | −0.02 | 0.417 |
| X4 | −0.498 | 0.759 | −0.243 | 0.313 |
| X5 | −0.281 | 0.512 | −0.788 | 0.049 |
| X6 | −0.419 | −0.418 | −0.635 | 0.388 |
| X7 | −0.259 | 0.884 | −0.076 | 0.203 |
| X8 | −0.493 | 0.815 | 0.301 | 0.034 |
| X9 | 0.016 | 0.987 | 0.105 | 0.12 |
| X10 | 0.912 | 0.284 | 0.003 | 0.292 |
| X11 | 0.967 | 0.099 | −0.034 | 0.231 |
| X12 | 0.95 | 0.045 | 0.004 | 0.152 |
| X13 | −0.73 | 0.13 | 0.614 | 0.258 |
| Contribution | 38.313 | 32.667 | 14.749 | 9.882 |
| Cumulative contribution | 38.313 | 70.98 | 85.729 | 95.557 |

**Note:**
X1, Bulk density; X2, Moisture content; X3, Capillary water holding capacity; X4, Maximum water holding capacity; X5, Minimum water holding capacity; X6, Capillary porosity; X7, Non-capillary porosity; X8, Total porosity; X9, pH; X10, Organic matter; X11, Total Nitrogen; X12, Total Phosphorus; X13, Total Potassium.

## Spatial heterogeneity of soil physical and chemical properties

The spatial variability of soil physicochemical properties at different vertical profile depths ranged from 21.91 to 87.59 m. The spatial autocorrelation variations of soil bulk density, soil moisture content, soil non-capillary porosity, soil capillary water holding capacity, and maximum water holding capacity under different depths of the soil layer were large (Fig. 2). The optimal semi-variance function fitting models for soil physical and chemical properties mainly include exponential model, spherical model, and Gaussian model. The abutment ratio Co/Co+C of soil physical and chemical properties ranged from 12.99% to 89.53%, in which the spatial dependence of pH, total phosphorus, and total potassium was very high in layer A (0–15 cm), and the spatial dependence of total soil porosity and maximum water holding capacity in layer A (0–15 cm), organic matter in layer B (15–30 cm), and total nitrogen and minimum water holding capacity in layer C (30–50 cm) were weaker. dependence was weak.

## Characterization of the spatial distribution pattern of soil physical and chemical properties

In this study, the spatial distribution pattern of soil physical and chemical properties was mapped using kriging interpolation in Arcgis, and it was found that the indicators of soil physical and chemical properties were spatially distributed with heterogeneous patches of different sizes and shapes. The degree of rocky desertification is an important factor affecting the spatial distribution pattern of soil physical and chemical properties. Therefore, in the southeastern part of the study area, where the degree of rocky

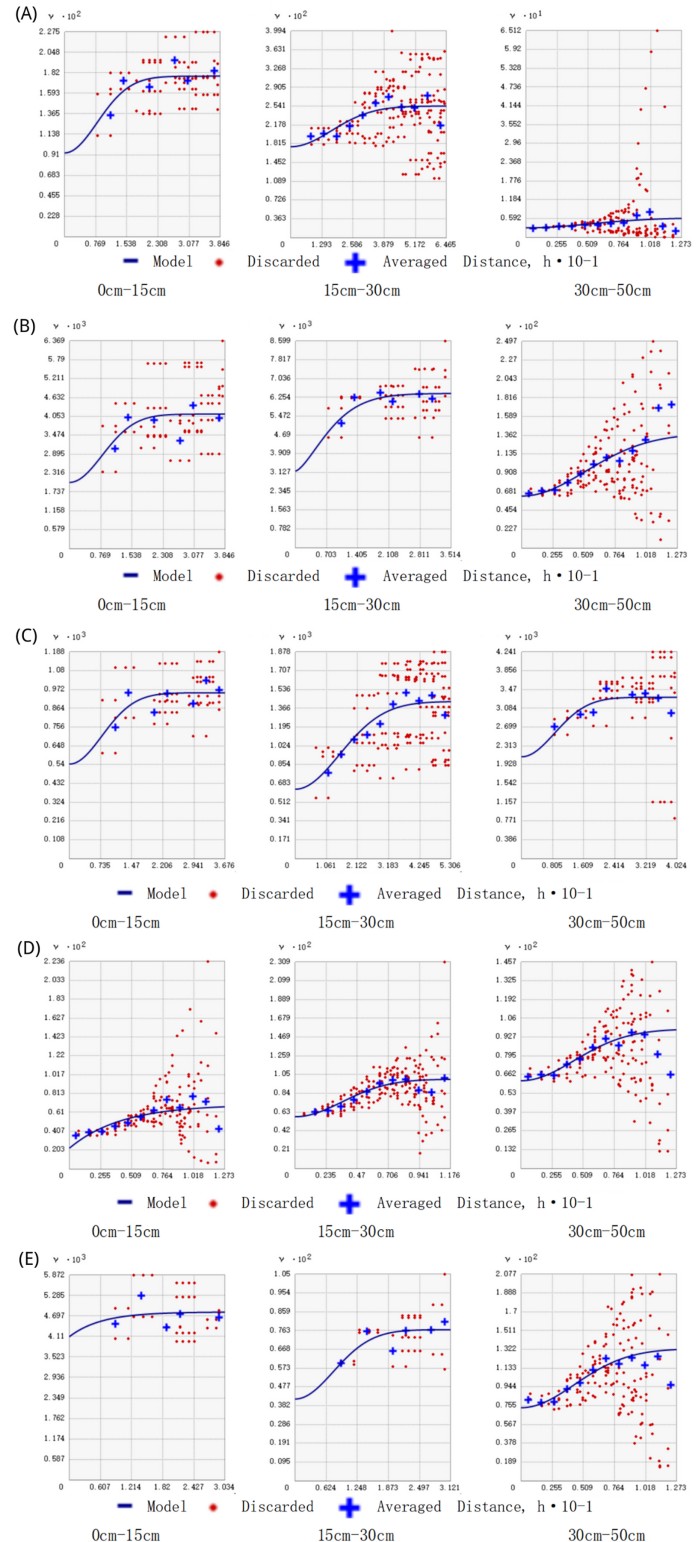

**Figure 2 Semivariance function model diagram.** (A) Soil bulk density for the three layers, (B) soil moisture content; (C) soil non-capillary porosity; (D) soil capillary water holding capacity; (E) maximum water holding capacity.

desertification is less, most of the indicators of soil physical and chemical properties appeared to have higher values. The spatial distribution structure of soil bulk density was highly variable, and the spatial distribution of soil capillary porosity, soil moisture content and total soil porosity were very similar (Fig. 3). The spatial distribution patterns of soils at different depths were characterized by their own features, but there was a certain degree of consistency.

## DISCUSSION

In this study, it was found that the soil capacity in the rocky desertification area was larger, and the mean value of water holding capacity was higher in the abandoned land because the terrain of the abandoned land was more gentle. This is consistent with the findings of *Yang et al. (2019)* in their study of the effect of slope on water flow rate. There was no significant difference in the non-capillary porosity between the two sites, which was consistent with the results obtained by *Sheng, Liu & Xiong (2013)* in their study of the response of soil physicochemical properties in the process of karst desertification succession in southern China. The soil nutrients organic matter, total phosphorus and total nitrogen contents were higher in the abandoned land. This is because of the high abundance of shrubs and grasses in the abandoned land, which is consistent with the results obtained by *Si et al. (2009)* in their study of the effect of vegetation on soil physicochemical properties in karst areas. Soil potassium content was higher within the rocky desertification area, which is consistent with the findings of *Li et al. (2015)*, who showed that the higher the degree of rocky desertification, the lower the potassium content. It was found that soil nutrients showed a decreasing trend with increasing soil depth in the artificial mixed forest and abandoned land. The organic matter content mainly depends on the process of humus leaching and infiltration on the surface (*Wu, He & Zhou, 2010*; *Wang et al., 2016*). This finding is in agreement with *Jing et al. (2016)* in their study. Soil organic matter is an important source of soil nitrogen and phosphorus nutrients (*Li et al., 2008b*; *Wang et al., 2019*). The main source of potassium is soil potassium minerals, and the content of total potassium in soil is affected by several aspects (*Guo et al., 2016*), and the soil nutrient elements change dynamically under the influence of natural and anthropogenic factors. Therefore, the content of nitrogen and phosphorus in soil can be increased by artificial fertilization.

Most previous studies have focused on the spatial heterogeneity of soil nutrients, suggesting that the spatial variability of soil physical properties is relatively low (*Šimečková et al., 2020*; *Ba et al., 2022*; *Sun et al., 2022*). However, this study found significant differences in the spatial autocorrelation ranges of several important physical properties, such as soil bulk density, soil moisture, and soil porosity, within the study area. For soil physical and chemical properties with large spatial autocorrelation variable range, one scholar found that the variable ranges of soil physical properties at 0–10 cm, 11–13 cm and 10–20 cm were different in his study of the spatial heterogeneity of soil physical properties in broad-leaved red pine forests in Northeast China (*Wang, 1999*). For agroecosystems, *Tsegaye & Hill (1998)* showed that the spatial autocorrelation variance of soil bulk density at 0–15 cm was 22.04 m, while in this study the spatial autocorrelation variance of soil bulk

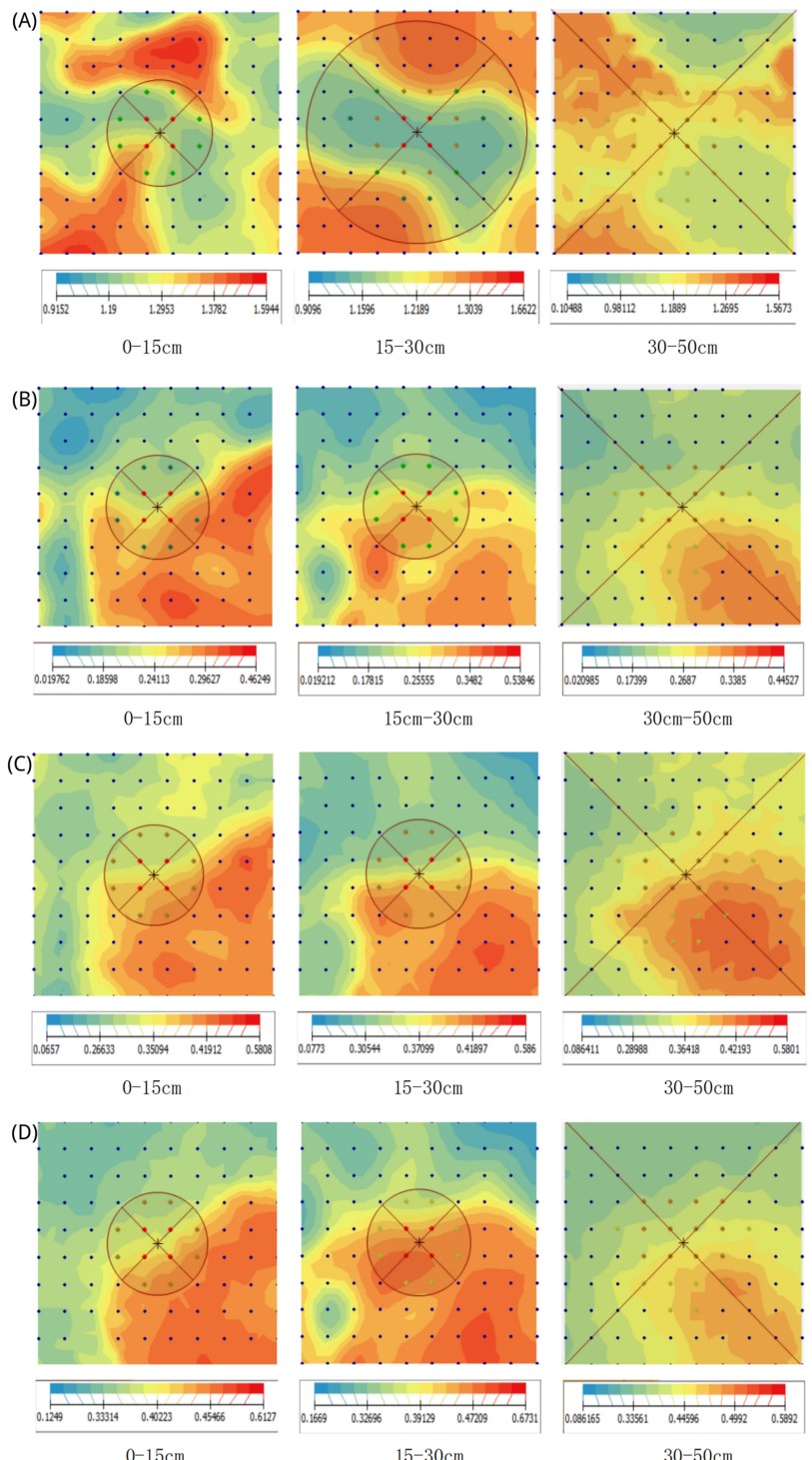

**Figure 3 The spatial distribution pattern.** (A) Soil bulk density for the three layers, (B) soil moisture content; (C) soil capillary porosity structure; (D) soil total porosity structure.

density at the same depth was 37.08 m, which may be due to a higher degree of spatial variability in the physical properties of the agro-systems under the influence of agricultural cultivation. In the study of spatial heterogeneity of soil in tropical rainforest ecosystems, one scholar found that the range of spatial autocorrelation of soil physical and chemical properties ranged from 10.81 to 48.65 m, while in the present study, the range of spatial autocorrelation of soil physical and chemical properties ranged from 21.91 to 87.59 m (*Shi, 2013*). In conclusion, the spatial heterogeneity of soil physical and chemical properties was found to vary across different ecosystems. *Iqbal et al. (2005)* studied a 162 hm$^2$ area and found that the range of topsoil bulk density was 106 m, which was much larger than the value of top soil bulk density range in the present study. It can be seen that the size of the research sample scale is an important factor in determining the distribution characteristics of soil physicochemical properties, and the ecological processes affecting soil physicochemical properties are completely different at different sample scales.

In addition, the spatial distribution pattern of soil physicochemical properties indicates that the degree of rocky desertification is a key factor affecting the spatial structure of these properties. The findings of this study are not only significant for understanding the dynamic changes of soil within ecosystems but also provide practical guidance for land management in rocky desertification areas, highlighting the importance of effective land management and restoration measures in these regions. Future research should further explore the factors influencing the spatial heterogeneity of soil physicochemical properties related to the degree of rocky desertification, in order to advance the development of this field and provide theoretical support for improving the ecological environment and sustainable land management. However, this study also has some limitations, such as a limited sampling time frame, which may not capture potential seasonal variations that could affect the comprehensiveness of the research results. Therefore, future studies should consider sampling in different seasons to obtain more comprehensive soil characteristic data.

## CONCLUSIONS

This study investigates the spatial variability of soil physicochemical properties at different depths (0–15 cm, 15–30 cm, and 30–50 cm) in artificial mixed forests in the rocky desertification area of Wuling Mountain. The results reveal distinct vertical variations in soil properties, with significant differences observed among various depths. The soil characteristics exhibited high spatial heterogeneity, with a coefficient of variation ranging from 0.1 to 0.74. Notably, the correlation between organic matter and total nitrogen content was the strongest. Principal component analysis revealed that four principal components explained 95.56% of the variance in soil properties, highlighting the dominant factors influencing soil characteristics in the study area.

The spatial variability in soil properties was found to range from 21.91% to 87.59%, with significant variations in key indicators such as bulk density, moisture content, and water-holding capacity across different soil horizons. The pH, total phosphorus, and total potassium in the A-layer (0–15 cm) exhibited high spatial dependence. Additionally, the degree of rocky desertification significantly influences the spatial distribution of soil

physicochemical properties. In the southeastern area, with lower desertification, most properties had higher values. While soils at different depths show distinct patterns, there is some consistency across them.

These findings provide important insights into the spatial patterns of soil properties under varying degrees of rocky desertification, with practical implications for land management. Understanding the soil characteristics at different depths can inform targeted soil improvement strategies, optimize water resource management, and aid in the restoration of degraded ecosystems. However, the study's limitations, such as the narrow sampling period and potential seasonal variations, suggest the need for further research. Future studies should increase sampling frequency and explore seasonal changes in soil properties to provide more comprehensive recommendations for land management.

### Funding

This work was supported by the Hunan Forestry Bureau (XLKY202330). The funders had no role in study design, data collection and analysis, decision to publish, or preparation of the manuscript.

### Grant Disclosures

The following grant information was disclosed by the authors:
Hunan Forestry Bureau: XLKY202330.

### Competing Interests

The authors declare that they have no competing interests.

### Author Contributions

- Ziqian Pan conceived and designed the experiments, performed the experiments, analyzed the data, prepared figures and/or tables, and approved the final draft.
- Yanyan Dong conceived and designed the experiments, performed the experiments, analyzed the data, prepared figures and/or tables, and approved the final draft.
- Gongxiu He conceived and designed the experiments, authored or reviewed drafts of the article, and approved the final draft.
- Tongtong Guo conceived and designed the experiments, performed the experiments, analyzed the data, prepared figures and/or tables, and approved the final draft.
- Ninghua Zhu conceived and designed the experiments, authored or reviewed drafts of the article, and approved the final draft.

### Data Availability

Raw data are available at Figshare:

Pan, Ziqian (2024). Spatial heterogeneity of soil properties in planted mixed forests in the rocky desertification areas of the Wuling Mountain. figshare. Media. https://doi.org/10.6084/m9.figshare.26356699.v2.
## Supplemental Information

Supplemental information for this article can be found online at http://dx.doi.org/10.7717/peerj.18724#supplemental-information.

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
