# Peer review of "Spatial heterogeneity of soil properties in planted mixed forests in the rocky desertification areas of the Wuling Mountain"

_PeerJ, doi:10.7717/peerj.18724_

## Round 0.1 · original submission · Major Revisions

· Academic Editor

Major Revisions

Dear Authors,

Reviewers have found serious issues in your manuscript. You are advised to address these issues and improve the quality of your manuscript, which will be re-reviewed.

Reviewer 1 ·

Basic reporting

Line 128 & 226, 244, Error! Reference source not found.?
Line 134-135 not clear. ‘About 500g of soil was taken from each layer and put into a cloth bag to pick out the debris in time.’
Line 135-137, And the natural environment of the sampling site was carefully recorded, including the presence of fallen trees and dead standing trees at the edge of the sampling site.
Sentence should not start with ‘And’
There is room to improve English language and its expression.
Line 140-144, rewrite
Analysis of soil should be cited with references.
Line 163-170, too long sentence, rewrite for 2-3 short sentences
In P<0.01, P need to be italicized in all MS
Discussion section need to be revisited. Preferably first discuss your own study results then argue
Lot of formatting issues need to be addressed in the whole MS, as for example, give a space before and after ( ).
Mistakes in referencing e.g., Iqbal J, Thomasson JA, Jenkins JN, Owens PR, Whisler FD. 2005. Spatial Variability Analysis of Soil Physical Properties of Alluvial Soils. Soil Science Society of America Journal 69. DOI: 10.2136/sssaj2004.0154.
Need to be corrected. More mistakes were observed
All figure captions/descriptions need to be elaborative and self explanatory.

Experimental design

The MS falls within the scope of PEERJ. Research gap well established.
Methods need to be described with appropriate details with references especially soil sample analysis.

Validity of the findings

Raw values of all soil analyses from which the means where calculated and used for statistical analyses need to be provided as raw data. There is no indication of replicates used to calculate mean values. How many soil samples /layer of soil? Raw data is not complete.
Conclusion section need to be revisited as methodology tone should be avoided

Reviewer 2 ·

Basic reporting

no comment

Experimental design

no comment

Validity of the findings

no comment

Additional comments

The article " Spatial heterogeneity of soil properties in planted mixed forests in the rocky desertification areas of the Wuling Mountain. In recent times, there is a great deal of interest in understanding the impact of regional ecological environment upon soil properties. This manuscript want to provide an insight into the understanding of physicochemical indexes and explore the nature and spatial heterogeneity of the soil of the planted mixed forests within the rocky desertification area of the Wuling Mountain. However, the research question posed in your manuscript was deemed not to be meaningful or significant within the broader context of the field. Additionally, the findings and conclusions presented did not provide sufficient new insights or useful information that would advance the understanding of the topic or contribute to the development of the field.

·

Basic reporting

Clarity of language: The manuscript is generally well-written, but there are instances where the English could be improved to ensure clarity. Certain sentences are unnecessarily complex and could be streamlined. Examples include:

Sentence in line 38: "The karst region desertification has appeared in the ecological environment has been infringed upon and led to the fragility of the ecosystem" – this is somewhat convoluted.
The phrase "appeared in the ecological environment" could be rephrased for clarity.
Introduction and Background: The introduction provides a good overview of the issue of rocky desertification and its ecological impacts, particularly in the Wuling Mountain area. The references used are relevant, but the manuscript could benefit from a more detailed explanation of why the specific depths (0-15cm, 15-30cm, 30-50cm) were chosen for soil sampling. This would help contextualize the study better.

Figures and Tables: The figures and tables are well-labeled and relevant to the study, but some require clearer descriptions. For example:

Figure 2 (Semivariance function model diagram) is difficult to interpret without better explanation.
The legends in the tables should include more detail about the metrics being presented, especially for readers unfamiliar with certain soil property terms.
Literature references: The manuscript is well-referenced with relevant sources, though it would benefit from more recent studies (most references are from early 2000s or before). Including more recent work on soil heterogeneity or karst ecosystems would strengthen the literature review.

Experimental design

Research scope and relevance: The study falls within the scope of PeerJ and addresses an important issue related to soil heterogeneity in rocky desertification areas. The research question is clearly defined and fills a knowledge gap, especially concerning soil spatial variability in mixed planted forests. The research could provide useful insights into ecosystem management.

Methods: The methods are generally well-detailed and appropriate for the study, though some aspects could use additional clarification:

The process of sampling and data collection is explained, but the reasoning behind the specific spatial intervals (21.91 m to 87.59 m) used in the geostatistical analysis could be better justified.
The authors should expand on how they handled potential sampling biases (e.g., edge effects, site disturbances).
More information on why kriging interpolation was chosen over other spatial interpolation methods would enhance the methodological transparency.

Validity of the findings

Data availability and analysis: The data presented is robust and generally supports the conclusions. The use of correlation, ANOVA, and principal component analysis is appropriate, though the interpretation of some statistical results could be improved. For instance:

The high variance contribution (95.557%) in the principal component analysis suggests that most of the variability in soil properties is explained by the components, but the implications of this high value could be discussed in greater detail.
The relationship between total nitrogen and soil organic matter (correlation coefficient of 0.947) is discussed, but the ecological implications of this relationship are not fully explored.
Conclusions: The conclusions are consistent with the findings and address the original research question. However, the manuscript would benefit from a more explicit link between the results and practical applications for land management in rocky desertification areas. The authors should also discuss the potential limitations of the study more thoroughly, such as the limited sampling time frame or potential seasonal variations.

Additional comments

General Comments:
Strengths: The manuscript offers a comprehensive study of soil heterogeneity in a key ecological region, using a variety of statistical and geostatistical methods. The findings regarding the relationship between soil organic matter and nitrogen, as well as the spatial variability of other key soil properties, are particularly valuable for land management and conservation efforts in rocky desertification areas.

Weaknesses: While the study is strong overall, it suffers from a lack of depth in discussing the broader ecological implications of the findings. Furthermore, the writing could be improved to make the manuscript more accessible to a broader audience, especially those who may not be familiar with specific soil science terms or geostatistical methods. Finally, the manuscript could benefit from more recent references to strengthen the literature review.

Reviewer 4 ·

Basic reporting

The primary issue with this study lies in its lack of innovation, as most of the information provided is descriptive and the authors have failed to present their results logically.

Additionally, a significant concern arises regarding the authorship list in this manuscript. Upon investigation, it was discovered that this manuscript is based on a master thesis published in 2021 by Yan-yan DONG (10.27662/d.cnki.gznlc.2021.000344) without proper acknowledgement or permission from Dong. Interestingly, one of the authors mentioned in this manuscript, Tongtong GUO, is also acknowledged separately in the Acknowledgements section. This raises potential conflicts that require clarification from the authors."

Experimental design

I had a quick review, please see comments maybe helpful to improve it:

Abstract:

It’s not clear the background and why it’s important to do such a study.

L20-21: I don’t understand what is a certain degree “Additionally, there was a certain degree of correlation between soils at different vertical profile depths”

Keywords:

soil should be Soil

Order in alphabetical

Introduction:

L60-61: references needed
L62: “degree of rocky desertification” is not a common use
L128: what’s wrong here ”(Error! Reference source not found.)”

L107: “Survey methodology” is not correct, you included lab analysis.
L143: total nitrogen for what? Content? Concentration?
L156: what is the difference between h and h

L197: there is no need to keep three digits in the text, see also in L207 -216.
L203-205: what is point for this paragraph?
L226-227: again, ”(Error! Reference source not found.)”
L236-246: I believe this paragraph belongs to Discussion

Discussion:

L249: What is the point to define SOIL here: “Soil is a fragile thin layer with species life activities and pore structure developed from rocks
through weathering (Miao, 2007).” ?

L250-252: “The soil in the sample area of Wuling Mountain National Rocky Desertification Comprehensive Treatment Research Base is limestone yellow-red soil. The soil has poor continuity and is patchily distributed in karst gullies and fissures”. This should be in the Introduction.

Validity of the findings

I had a quick review, please see comments maybe helpful to improve it:

Abstract:

It’s not clear the background and why it’s important to do such a study.

L20-21: I don’t understand what is a certain degree “Additionally, there was a certain degree of correlation between soils at different vertical profile depths”

Keywords:

soil should be Soil

Order in alphabetical

Introduction:

L60-61: references needed
L62: “degree of rocky desertification” is not a common use
L128: what’s wrong here ”(Error! Reference source not found.)”

L107: “Survey methodology” is not correct, you included lab analysis.
L143: total nitrogen for what? Content? Concentration?
L156: what is the difference between h and h

L197: there is no need to keep three digits in the text, see also in L207 -216.
L203-205: what is point for this paragraph?
L226-227: again, ”(Error! Reference source not found.)”
L236-246: I believe this paragraph belongs to Discussion

Discussion:

L249: What is the point to define SOIL here: “Soil is a fragile thin layer with species life activities and pore structure developed from rocks
through weathering (Miao, 2007).” ?

L250-252: “The soil in the sample area of Wuling Mountain National Rocky Desertification Comprehensive Treatment Research Base is limestone yellow-red soil. The soil has poor continuity and is patchily distributed in karst gullies and fissures”. This should be in the Introduction.

---

## Round 0.2 · Minor Revisions

· Academic Editor

Minor Revisions

Please address the remaining comments. Please improve the language and figure quality of your manuscript.

Reviewer 1 ·

Basic reporting

The English grammar as well as the expression of English need to be improved in the whole MS. e.g., Line 139-140 Take about 500 g of soil from each layer and put it in a ziplock bag, picking out debris such as plant roots, stones, dead branches and leaves in time.
Similarly, line 18-=183, Soil physical and chemical properties changed regularly with the increase of soil depth. As depth increases, soil bulk density gradually rises, while soil moisture content, porosity, and water-holding capacity generally decline. The contents of organic matter, TN, and TP decrease significantly, especially organic matter, while TK increases with depth.
& lot more…
Lot of formatting issues still need to be addressed in the whole MS.
All figure quality needs to be improved.

Experimental design

The authors addressed the previous comments

Validity of the findings

Conclusion section need to be revisited as methodology tone should be avoided (although pointed out earlier) and length be reduced keeping only main findings.

·

Basic reporting

The manuscript is well-structured, with a clear objective focused on the spatial heterogeneity of soil properties in mixed forests in rocky desertification areas of the Wuling Mountain. The authors have addressed reviewers' suggestions, including enhancing clarity, improving language, and adding recent references. Formatting issues, such as figure captions and citation corrections, have been resolved, and the results are now presented in a logical sequence. The manuscript includes detailed explanations of soil physical and chemical characteristics, ensuring that the content is accessible to readers in the field of environmental science and soil ecology.

Experimental design

The study design is robust, aligning with the scope of the journal and addressing a critical ecological issue—soil variability in rocky desertification areas. The experimental methods are adequately described, with the soil sampling process and depth intervals clearly outlined. The authors have addressed potential sampling biases and chosen kriging interpolation for spatial analysis due to its suitability for this type of data, as suggested by reviewers. Statistical analyses, including principal component analysis (PCA) and analysis of variance (ANOVA), are appropriate for the dataset, and method references have been updated and expanded for clarity.

Validity of the findings

The findings are solid, supported by comprehensive data analysis that aligns with the study’s objectives. The authors have clarified statistical interpretations, particularly in PCA and correlation analysis, to provide ecological insights into the relationship between soil organic matter and nitrogen. Potential applications for land management in rocky desertification areas are well-discussed, along with limitations, such as seasonal sampling considerations. These revisions strengthen the ecological relevance and practical implications of the study's conclusions.

Additional comments

Dear Editor,

The authors have carefully addressed all suggestions and feedback provided by the reviewers, revising the manuscript to ensure accuracy, clarity, and compliance with formatting guidelines. The tracked changes version highlights all adjustments made to the language, structure, and content to enhance readability and logical flow. All references and citations were reviewed for accuracy, with any formatting errors corrected. Figure captions and tables have been improved for clarity, and additional recent studies were cited to strengthen the manuscript's contextual background. The authors believe the manuscript now meets the standards for publication and respectfully submit it for acceptance.

---

## Round 0.3 · Minor Revisions

· Academic Editor

Minor Revisions

Dear Authors,

Please remove typos and grammatical mistakes from your manuscript as recommended by the reviewer. For your ease, please make corrections in the following sentences:
L10, L21, L22, L41, L73, L74, L76, L79, L91, L99, L169, L185, L189, L302, L303, L326

Reviewer 1 ·

Basic reporting

The authors did not address my previous comments.
1- Professional English editing recommended.

Experimental design

-

Validity of the findings

The authors did not address my previous comments.

---

## Round 0.4 · accepted · Accept

· Academic Editor

Accept

I confirm that the authors have addressed all of the reviewers' comments. I am happy with the current version. This manuscript is ready for publication.